# Agrometeorological and Morpho-Physiological Studies of the Response of Red Currant to Abiotic Stresses

Olga Panfilova [1,*], Mikhail Tsoy [1], Olga Golyaeva [1], Sergey Knyazev [1] and Mikhail Karpukhin [2]

[1] Russian Research Institute of Fruit Crop Breeding (VNIISPK), Zhilina, Orel District, 302530 Orel Region, Russia; nauka@vniispk.ru (M.T.); golyaeva@vniispk.ru (O.G.); info@vniispk.ru (S.K.)
[2] Ural State Agrarian University, ul. Karl Liebknecht, 42, 620075 Yekaterinburg, Russia; mkarpukhin@yandex.ru
* Correspondence: us@vniispk.ru

**Abstract:** The aim of this work was to study the mechanism of climatic adaptation of red currant genotypes (*Ribes rubrum* L.) on the basis of physiological, biochemical and agrometeorological measurements and to determine the different phenophases of plant development identify adaptive genotypes for introduction. The studies were carried out in 2014–2017. The indicators of the water status of annual shoots (water content, water retention capacity), the biochemical composition of berries (vitamin C) and phenological observations were evaluated, taking into account meteorological data. The genotypes of *R. petraeum* Wulf. and *R. multiflorum* Kit. had the longest production period. Ambiguous data on the influence of temperature on the content of ascorbic acid in berries were revealed. High temperatures (>+26 °C) contributed to a greater accumulation of ascorbic acid in the cultivars of *R. vulgare* Lam. High accumulations of vitamin C in the range of +25–27 °C were found in *R. petraeum* Wulf. and *R. multiflorum* Kit.. High water content and water loss contributed to early recovery from the dormant state and reduced resistance to spring temperature changes in *R. vulgare* Lam. Genotypes of *R. vulgare* Lam., and *R. multiflorum* Kit. are promising for growing in a zone with a temperate continental climate. The genotypes of the species *R. petraeum* Wulf are suitable for introduction to the areas with a continental climate. The obtained results are important for adaptive gardening.

**Keywords:** *Ribes rubrum* L.; phenological phases; ascorbic acid; water status; adaptation

## 1. Introduction

Over the past decade (2010–2020), there has been a positive trend in global demand for the production of berry crops, which is primarily due which is primarily due lower prices for consumers combined with popularization of certain benefits of berry consumption for human health, as well as to the improvement of new agronomic measures that have allowed the traditional cultivation areas to expand. Berry crops, as a rule, are productive, have a short period of entry into fruiting, and have a high nutritional value, so their production and sale can represent a certain positive contribution to the development of a country's economy [1,2]. Red currant (*Ribes rubrum* L.) is a promising berry in this context; its important biological features include precocity, productivity and high profitability. Red currant berries have excellent taste and medicinal qualities and are rich in physiologically active substances such as vitamins, microelements, organic acids, etc. [3,4]. A long period of red currant resistance to berry shedding increases the period of fresh fruit consumption significantly, and at the same time the taste and nutritional properties are preserved when berries are frozen. There is a therefore a high demand for red currant berries for use in processed products such as jams, pastes, juice, etc. [5]. The main production of red currant berries is concentrated in Germany, France, Poland, Great Britain, Russia and Ukraine [6,7].

Recently, frequent climatic anomalies have been the main reasons for the decline in adaptability and productivity of horticulture crops. This is very significant during

introduction of new genotypes. New measurements in fruit breeding play an important role in the creation of new highly adaptive genotypes allowing producers to make quick, accurate development forecasts under new environmental conditions. The physiological characteristics of plants and the biochemical composition of fruits are determined by their genetic origin and are influenced by natural and climatic factors. The water status of plants for example determines their level of adaptation to temperature stress. Previous studies on apple, grape, and currant cultivars growing in different natural and ecological zones have shown an increase in the content of bound water as a percentage of free water, which is due to an increase in the water content of the shoot cell plasma and the resistance of cell membranes to destruction during low-temperature stress [8–13]. However, the researchers did not take into account the specific origin of the genotypes of fruit and berry crops in the growing areas, nor the change in the state of the water status during all winter months to identify the period when plants were least affected by low-temperature stress, as well as to predict the general condition of plants after winter (damage to generative or flower buds, annual shoots) and yield.

Weather conditions have a certain effect on the biochemical composition of fruits. The content of ascorbic acid (vitamin C), phenolic compounds, and anthocyanins were higher in the years of optimal temperature (+24–+25 °C) [4,14–18]. However, the relationship of growing conditions with the genetic and geographical origin of fruit and berry cultivars was not taken into account and was not carried out, which is important for the promising direction in breeding, i.e., "obtaining new genotypes of horticulture crops with increased nutritional and biological value of fruits", and this is especially important for obtaining products of a certain functional purpose.

The purpose of the study is to evaluate red currant cultivars of different species origin according to a complex of adaptive-significant characteristics, to determine promising measurements for diagnosing the resistance of cultivars to climatic factors, as well as to identify the elements of adaptation of genotypes to stress factors of winter and spring periods in the central region of Russia and the prospects of introducing these cultivars to other climatic zones.

## 2. Materials and Methods

The work was carried out during winter (December–February), spring (March–May) and summer (June–July) 2014–2017. Russian Research Institute of Fruit Crop Breeding (VNIISPK) is located in the South-West of the Central region of Russia, 52°96′ north latitude; 36°07′ east longitude. The climate is temperate continental. The climate is temperate. The winter is moderately cold, with an average January temperature is −9 to −11 °C (16 to 12 °F). Summers are warm and humid, with an average July temperature is 19 to 21 °C (66 to 70 °F). Rainfall averages 520 to 630 mm (20 to 25 in), and snow cover averages 120 days. Plant samples were selected at the collection site of the red currant (Figure 1).

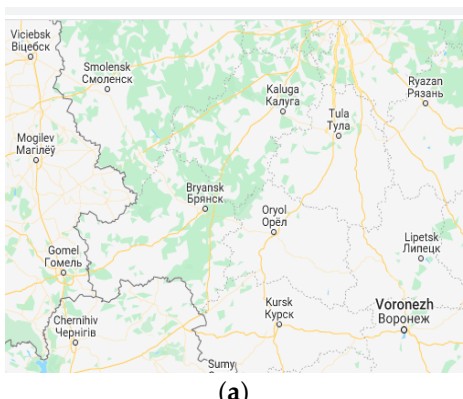
(**a**)

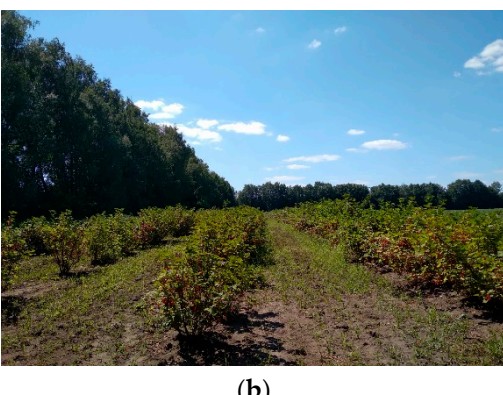
(**b**)

**Figure 1.** Geographical map of the Orel region (**a**) and genetic collection of red currant Russian Research Institute of Fruit Crop Breeding (**b**).

The planting distance was 2.8 × 0.5 m. The soils of the experimental plot were Loamy Haplic Luvisol, the capacity of the humus horizon is 30–55 cm, the humus content is 3–5%, pH is 4.72, the content of K and P exchange in the horizon of 0–20 cm is 16.36 and 19.12 mg/100 g of soil, respectively. To determine the amount of P and K in the soil, an extractant 0.2 n HCI was used. For the analysis 10 g of soil and 50 ml of 0.2 n HCI (1:5) were used. The object of the study was to evaluate 11 red currant genotypes (Table 1).

**Table 1.** The origin of red currant genotypes.

| Variety | Genetic Origin | Country Originator | Genetic Species | Geographic Origin |
|---|---|---|---|---|
| Jonkheer Van Tets | Faya Plodorodnaya × London Market | Holland | | |
| Niva | Minnisota × Chulkovskaya | Russia | | |
| Asya | Chulkovskaya × Maarses Prominent | Russia | *R. vulgare* Lam. | Forest zone of Eurasia and East Europe |
| Roza | Chulkovskaya × Rose Chair | Russia | | |
| Gazel | Chulkovskaya × Maarses Prominent | Russia | | |
| Belka | Chulkovskaya × Red Lake | Russia | | |
| Hollandische Rote | unknown | Holland | *R. petraeum* Wulf. | Mountains of Central and South Europe |
| Dana | Rote Spätlese × Jonkheer Van Tets | Russia | | |
| Dar Orla | Rote Spätlese × Jonkheer Van Tets | Russia | *R. multiflorum* Kit. | Mountain areas and slopes of South Europe |
| Podarok Let'a | Rote Spätlese × Jonkheer Van Tets | Russia | | |
| Osipovskaya | Rote Spätlese × Jonkheer Van Tets | Russia | | |

Meteorological data were recorded from 2014–2017, including daily maximum and minimum temperatures, precipitation as well as monthly averages were calculated, with which the studied traits were compared. According to the data of the meteorological station, the weather conditions of the winter periods did not vary greatly over the years of the research. Winter 2014–2015 was moderately cold with little snow cover. The average monthly temperatures of the winter months are shown in Table 2.

**Table 2.** Weather conditions in 2014–2017.

| Year | Month | $T_{average}$ °C | $T_{min}$ °C | $T_{max}$ °C | Year | Month | $Tt_{average}$ °C | $T_{min}$ °C | $T_{max}$ °C |
|---|---|---|---|---|---|---|---|---|---|
| 2014 | December | −12.1 | −18.8 | +1.2 | | May | +13.8 | +0.5 | +25.6 |
| | January | −5.4 | −24.5 | +4.2 | | June | +15.8 | +9.3 | +31.5 |
| | February | −4.8 | −17.4 | +4.5 | 2016 | July | +19.3 | +15.9 | +34.2 |
| | March | −1.1 | −7.5 | +14.5 | | December | −4.2 | −20.6 | +2.2 |
| 2015 | May | +12.3 | −1.2 | +23.3 | | January | −6.2 | −24.0 | +1.2 |
| | June | +14.6 | +5.7 | +30.8 | | February | −5.8 | −14.2 | +6.5 |
| | July | +18.1 | +12.3 | +29.7 | | March | +1.3 | −8.0 | +11.5 |
| | December | −9.3 | −10.0 | +9.2 | 2017 | May | +14.6 | −1.6 | +25.5 |
| | January | −5.2 | −29.3 | +3.2 | | June | +17.2 | +7.3 | +28.1 |
| 2016 | February | −4.9 | −12.4 | +2.1 | | July | +18.6 | +13.1 | +29.0 |
| | March | +0.4 | −11.0 | +10.3 | | December | −7.8 | −10.8 | +0.7 |

During the winter, there were sharp changes in the air temperature, and frosts were replaced by thaws. Rather frequent winter thaws were the cause of unstable snow cover

and led to an imbalance of the protective mechanisms in the plants. December and January 2015–2016, and 2016–2017 were quite cold. In December 2016 and 2017, there were frequent daytime thaws and low nighttime temperatures. Snow cover was established only by the third week of December. The average monthly temperature in January exceeded the long-term average ($-9.2$ °C) by 4 °C in 2016 and by 3 °C in 2017. The temperature of February did not vary much over the years of the research. In March, the snow melted quite quickly and there were sharp temperature changes that led to certain damage to plants. The weather conditions of the growing seasons were characterized by their non-alignment (Table 1). In some years, there were sharp changes in night and day temperatures, which led to partial death of generative buds. In some years, in the II–III weeks of June, air and soil droughts were observed, which led to a decrease in the yield and quality indicators of berries; so during the II weeks of June 2015 and the III weeks of 2016, high temperatures with a small amount of precipitation led to early shedding, and in some cases, wilting of berries. July was observed the hottest month in the study area. In 2014, 2015 and 2017, the temperature of this period did not deviate much from the average annual value, although it should be noted that during this month there were short-term dry periods of no more than 5–6 days, during which the temperature rose to $+33.0$–$+34.6$ °C. Such differences contributed to a faster and more harmonious ripening of berries and did not have a strong impact on the quality and quantity indicators of red currant berries. A prolonged drought was observed in 2016 during the whole month. That year, a high percentage of falling berries was recorded.

Annual phenological records (date/trait) were carried out the beginning of vegetation (or bud break)—the extension of the green cone of leaves in 10% of buds; the beginning of flowering—the blossoming of 3–5% of flowers on bushes; the beginning of ripening—when the first colored berries appear; full maturation when the berries are fully ripe and have a typical taste, color, aroma for the cultivar. Individual for cultivars belonging to different types of phenophase of development, temperature minima and sums of effective temperatures, regression coefficients, equations of the dependence of sums of average daily temperatures for the studied period ($\Sigma T$) on the duration of the period L were determined:

$$\Sigma T = A + BL, \tag{1}$$

where B is a constant value and is interpreted as the temperature minimum of the period; A is a sum of the effective temperatures.

The growing degree days (GDD) were determined from 2014 to 2017. The studies were determined at the Beginning of the growing season of currants, when the sum of effective temperatures is above $+5$ °C (April), also Beginning of flowering–full flowering (end of April–beginning of May) and Beginning of ripening–full ripening (June–July). The growing degree days was calculated using the following formula:

$$GDD = \frac{T_{max} + T_{min}}{2} - T_{base} \tag{2}$$

where $T_{max}$ is the maximum temperature, (°C); $T_{min}$ is the minimum temperature, (°C); $T_{base}$ is the effective temperature, for red currants this amount is $+5$ °C.

Biochemical studies, i.e., quantitative determination of vitamin C content, were carried out by the iodometric method: titration of ascorbic acid stained extracts (samples) with potassium iodate in an acidic medium in the presence of potassium iodide and starch. Starch formed with iodine a complex adsorption compound of a blue colour. The experiments were carried out in three biological repetitions.

The total amount of water in the annual shoots and the water content of the shoots were determined from December to March according to the method of Leonchenko et al. [19]. To determine the total water content, the shoots were crushed and weighed at a weight of 20 g on a Scout Pro SP 202 laboratory scale (OHAUS, Parsippany, NJ, USA). The samples were placed in metal containers and dried at a temperature of $+105$ °C for 5 h in an "Espec"

PSL-2KRN climate chamber (ESPEC, Osaka, Japan). The water content was calculated using the following formula:

$$K = \frac{m1 - m0}{m1} \cdot 100 \qquad (3)$$

where $m1$ is the initial weight of shoots, (g); $m0$—is the initial weight of shoots, (g).

The water content of the shoots was determined by the amount of water lost after 12 h of drying at a temperature of +25 °C. The water-retaining capacity of the shoots was calculated by the following formula:

$$W = \frac{m1 - m2}{m1 - m0} \cdot 100 \qquad (4)$$

where $m1$—is the initial weight of shoots, (g); $m2$—is the weight after wilting at a temperature of +25 °C, (g); $m0$—is the absolutely dry weight of the shoot after drying at a temperature of +105 °C, (g).

The field assessment of the condition of plants after overwintering was carried out visually in the field in the second weeks of April.

*Estimation of Low Temperature Damage*

Temperature damage of generative buds and vegetative shoots and general condition of plants were taken into account on the scale: 5.0 points—complete freezing of annual and perennial shoots; 4.0 points—weak condition of plants, more than 80% of buds and tissues were damaged; 3.0 points—average freezing, annual and perennial shoots and buds were damaged to 70%; 2.0 points—weak freezing: single damage of perennial shoots, annual shoots were damaged to 40%; 1.0 point—very weak freezing: $\frac{1}{4}$ of the upper part of the annual shoots damaged, perennial shoots were not damaged; 0 points—excellent condition, shoots and buds were not damaged [20].

The general condition of the plants was evaluated using a point scoring system: 5.0 points—excellent condition, healthy bushes with a large number of annual shoots, the leaf blade is well developed, there are no frost damage; 4.0 points—good condition, the bushes are healthy, have good foliage, have more than 10 annual shoots per bush, the leaves are not deformed, have minor up to 10% damage to the shoots by frost, diseases and pests, the plants are not in a depressed state; 3.0 points—average condition, the bushes are not very weakened (up to 25–30%) as a result of frost damage, diseases and pests, the leaves are not sufficiently developed, there are not many annual growth (4–5 annual shoots per bush); 2.0 points—weak condition, bushes are severely damaged by frosts, diseases and pests, low and weak development of annual growth (less than three annual shoots), leaves are of atypical shape and color, lag behind in the timing of the passage of phenophases (the period of the beginning of budding and flowering is very late); 1.0 point—plants are severely weakened, sick, do not have annual growth, badly damaged plants that are almost dying [20].

The dispersion analysis was carried out to identify significant differences between cultivars for the studied characteristics [21]. In the course of the study carried out for the correlation and regression analysis of the terms and duration of the phenoperiods between the different growth phases of development and weather was carried out. Regression equations were constructed using the Microsoft Office Excel 2010 software package, and Stat Soft Statistica 6.0 with sequential inclusion of variables. As for indicators, the dates of the steady transition of mean temperature were considered through the following limits: 0 °C, 5 °C, 10 °C, 15 °C, 20 °C and 25 °C, and through the number of days between them and the sum of temperatures and precipitations. The nonparametric Manna-Whitney criterion was used to compare the average values for the groups of cultivars [22]. Spearman's rank correlation coefficient was used to identify the duration between the relationship of the studied traits. The significance level of 5% was accepted in the study.

## 3. Results and Discussion

### 3.1. Agrometeorological Measurements of Prediction of Phenological Phases

The passage of phenological cycles in red currants is influenced by seasonal changes in the environment and, above all, the temperature factor. Several researchers studying the phenorhythms of berry and fruit crops have confirmed the high dependence of the beginning of bud break on night and day temperatures [23–25]. For many cultivated plants the daily temperature is +4.4–+5.0 °C. Tooke and Battey [26] also noted the dependence of the start date of flowering on growing conditions, temperature and humidity. Roetzer and Chmielewski [27] reported that the duration of phenological phases is also determined by the sum of active and effective temperatures, this effect varies among cultivars and plant species. To determine the correspondence of climatic factors (primarily temperature) to the south-west part of the central region of Russia and to predict the passage of the main phenological phases of red currant development (the beginning of vegetation, flowering, fruiting), the growing degree days (GDD) criterion was used. In this experiment, the beginning of vegetation in all red currant cultivars of the VNIISPK gene pool occurs when the sum of effective temperatures above +5 °C is reached.

This means that, for red currant, all time below 5 °C doesn't "count" in development. Table 3 shows that the period of the onset of vegetation in currants varied by years and did not depend on the cultivar. For all of the studied cultivars, the GDD measure was the same. A high dependence of the beginning of vegetation on the temperature factor was revealed (r = +0.68).

The average duration of the production period of the studied red currant cultivars from the beginning of bud break to full fruiting varies from 75 days for early and medium-early cultivars of *R. vulgare* species up to 91 days for late cultivars of *R. petraeum* and *R. multiflorum* species.

The clearest differences in the passage of the phenophases of the studied cultivars were noted at the stage of flowering and the beginning of maturation. The average duration of the period from the beginning of the vegetation to the beginning of flowering in the cultivars belonging to different species was from 19 to 25 days (r = + 0.47), from the beginning of flowering to the beginning of ripening of berries—50–61 days (r = +0.77) and from the beginning of ripening to full maturity of berries—6–7 days (r = +0.65).

The beginning of flowering occurs on average at a temperature above +16 °C (+16–+17.6 °C for 2014–2017). This date correlates with the date of transition of temperatures through +15 °C, the average correlation coefficient for the studied cultivars is r = + 0.76. The equation of regression of the date of the beginning of flowering from the date of temperature transition above +15 °C was: $y1 = 0.79 \times 15 + 6.6671$. The day's delay of the temperature transition above +15 °C results in a delay of the beginning of flowering for 0.8 day.

The date of full ripening of berries depends on the date of the beginning of ripening and high temperatures above +20 °C for the cultivars of early ripening and above +25 °C for later ripening cultivars. The average date of ripening of red currant berries was described by the equation: $y2 = 0.29 \times 20 + 15.4767$. The current study, the beginning and full ripening of berries occur almost after a constant number of days for the cultivar, accelerated by sufficiently high temperatures.

The obtained GDD data and correlation analysis confirmed that the natural and climatic factors of the south-west part of the central region of Russia correspond to the optimal level of passing the phenological stages of red currant development. Differences in the red currant cultivars at the stage of flowering and maturation were determined primarily by genetic origin. In accordance with the international classifier [28], 11 studied cultivars are classified according to the maturation periods as early—'Asya' and 'Niva', medium-early—'Gazel' and 'Jonkeer van Tets', medium—Roza, Belka and late—Dana, 'Dar Orla','Hollandische Rote', 'Osipovskaya' and 'Podarok Leta'.

**Table 3.** Passage of phenological phases of red currant cultivars. *R. vulgare* Lam.—cultivar of early and mid-early ripening; *R. petraeum* Wulf.—cultivar of medium-late ripening; *R. multiflorum* Kit.—cultivar of late ripening, 2014–2017.

| Variety | Beginning of the Growing Season | | Beginning of Flowering | | Beginning of Ripening-Full Ripening | |
|---|---|---|---|---|---|---|
| | Dates, Month | Growing Degree Days (GDD) | Dates, Month | Growing Degree Days (GDD) | Dates, Month | Growing Degree Days (GDD) |
| *R. vulgare* Lam. | | | | | | |
| Niva | 11.04–16.04 | 5–7 | 26.04–4.05 | 57–86 | 14–26.06 | 420–630 |
| Asya | 11.04–16.04 | 5–7 | 26.04–4.05 | 57–87 | 16–28.06 | 394–624 |
| Gazel | 11.04–16.04 | 5–7 | 28.04–6.05 | 69–106 | 24–30.06 | 459–641 |
| Belka | 11.04–16.04 | 5–7 | 28.04–6.05 | 69–102 | 24.06–1.07 | 451–644 |
| Roza | 11.04–16.04 | 5–7 | 28.04–6.05 | 69–118 | 24.06–3.07 | 467–644 |
| Jonkheer Van Tets | 11.04–16.04 | 5–7 | 29.04–5.05 | 76–112 | 18.06–30.06 | 460–629 |
| *R. petraeum* Wulf. | | | | | | |
| Hollandische Rote | 11.04–16.04 | 5–7 | 30.04–7.05 | 84–118 | 24.06–3.07 | 535–674 |
| *R. multiflorum* Kit. | | | | | | |
| Dana | 11.04–16.04 | 5–7 | 30.04–7.05 | 84–145 | 28.06–12.07 | 623–748 |
| Osipovskaya | 11.04–16.04 | 5–7 | 1.05–9.05 | 93–138 | 24.06–10.07 | 542–742 |
| Dar Orla | 11.04–16.04 | 5–7 | 1.05–8.05 | 93–138 | 30.06–12.07 | 631–742 |
| Podarok Leta | 11.04–16.04 | 5–7 | 1.05–9.05 | 93–145 | 30.06–14.07 | 641–777 |

### 3.2. Ascorbic Acid Content

Significant varietal differences in the content of vitamin C in red currant berries were revealed. For 2014–2017 the vitamin C levels varied from 25.4 to 49.56 mg 100 $g^{-1}$. A high content of vitamin C was typical for 'Niva' (50 mg 100 $g^{-1}$) and 'Jonkeer van Tets' (42 mg 100 $g^{-1}$) which are *R. vulgare* species, and the coefficient of the variation of that trait in these cultivars (V) was in the range of 16–18%. The vitamin C content trait showed variable values by years. Ambiguous data on the influence of the temperature factor on the ascorbic acid content of the representatives of three species of *Ribesia* were obtained. In the *R. vulgare* cultivars, a direct dependence of the vitamin C accumulation on the temperature was revealed ($R^2$ = +0.64). The current study reported a high correlation of vitamin C accumulation in red currant berries under the influence of high air temperatures during the harvest season. (Figure 2a).

Similarly, a high correlation of vitamin C accumulation in black currant berries under the influence of high air temperatures during the harvest season has been reported earlier [29,30].

However, few previous studies reported contradictory results on the destructive effect of high temperatures on the synthesis and accumulation of ascorbic acid and considered moderate air temperatures to be the most optimal [31–34]. In the present experiment, a similar dependence was noted for late-ripening *R. petraeum* Wulf. and *R. multiflorum* Kit. Cultivars. Simultaneously, a high inverse relationship between the studied trait and the temperature factor was noted in the representatives of *R. petroleum* and *R. multiflorum* ($R^2$ = +0.84). High average temperatures led to a decrease in the synthesis and accumulation of ascorbic acid in the berries of late ripening red currant cultivars (Figure 2b). During the four years of the research, in the cultivars of this group there was a high negative relationship of high temperatures with the content of vitamin C. Its maximum accumulation and content were determined by a temperature range of +25–+27 °C.

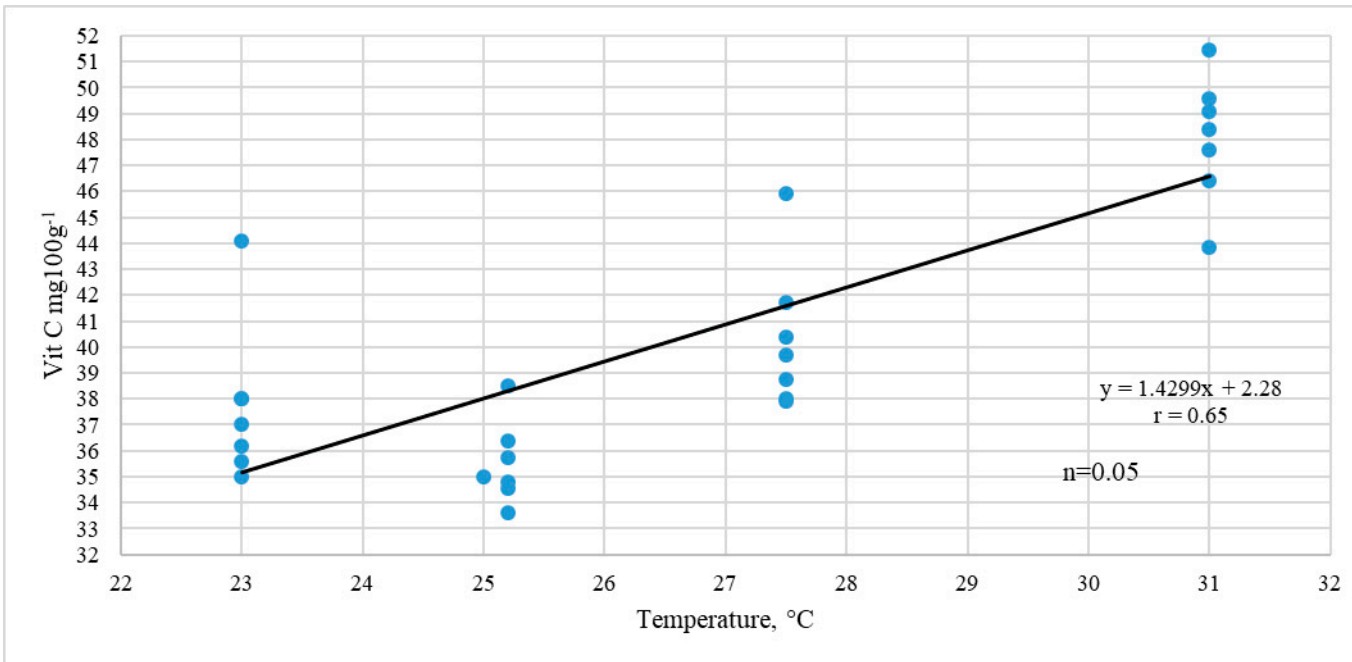

(**a**)

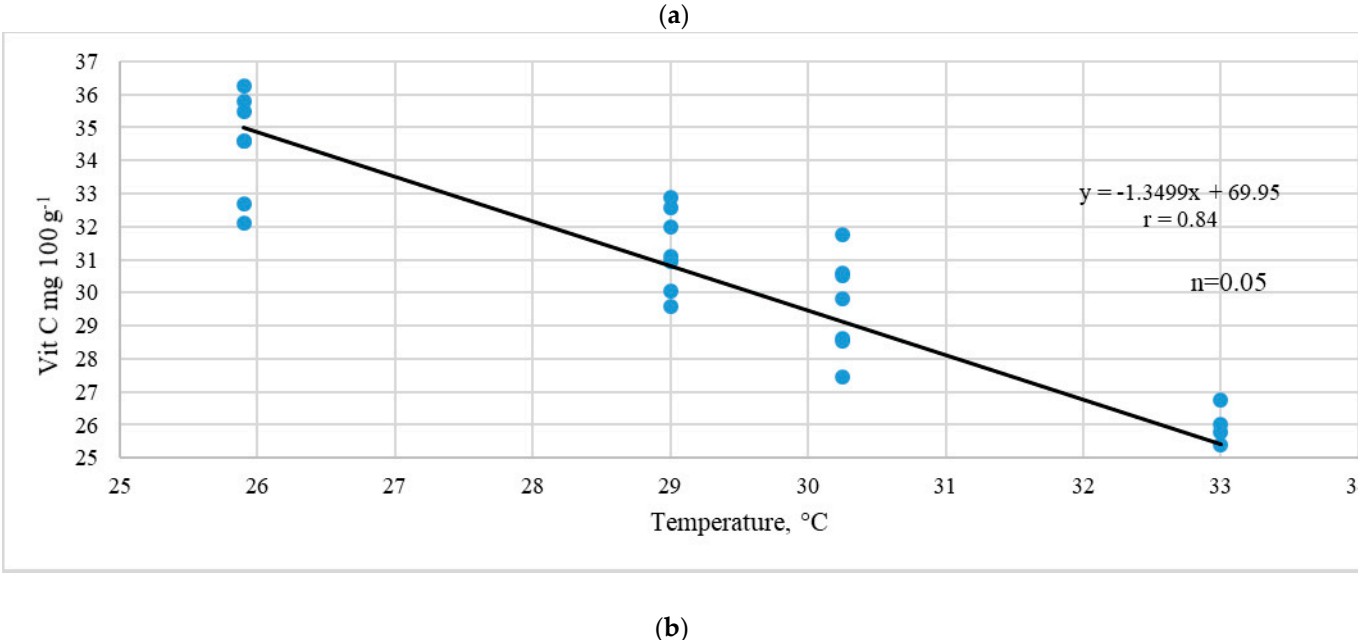

(**b**)

**Figure 2.** The relationship between the average temperature and the content of vitamin C in red currant berries of the species *R. vulgare* Lam. (**a**) and *R. petraeum* Wulf. and *R. multiflorum* Kit. (**b**) in 2014–2017. r—correlation coefficient, n—the level of significance of the trait.

### 3.3. Winter Hardiness of the Cultivars

The success of promoting new cultivars to other climatic zones (introduction) is determined by their adaptability to new growing conditions [35]. One of the important restraining factors of the distribution of red currant cultivars to new regions is associated with their resistance to early autumn frosts, low negative temperatures of the winter period, as well as to early spring frosts, even though red currant is considered to be a quite hardy berry culture, it is susceptible to sharp changes in night and day temperatures of the winter and thaws of the early spring periods [36–38]. This leads to a deterioration of the overall condition of the plant and a decrease in the economic yield. To identify resistant red currant

cultivars, the water content in annual shoots and the degree of its consumption during the winter months of the region were studied. The present study showed the variability of the traits under observation by the months during the research. At the beginning of winter (December), there were quite sharp temperature changes against the background of that, which led to the increased imbalance of energy systems of currant plants. And already at this stage, there was a different degree of preparation of cultivars for the dormancy period. The *R. petraeum* Wulf. cultivars were characterized by the best indicators of adaptation to temperature changes (Figure 3).

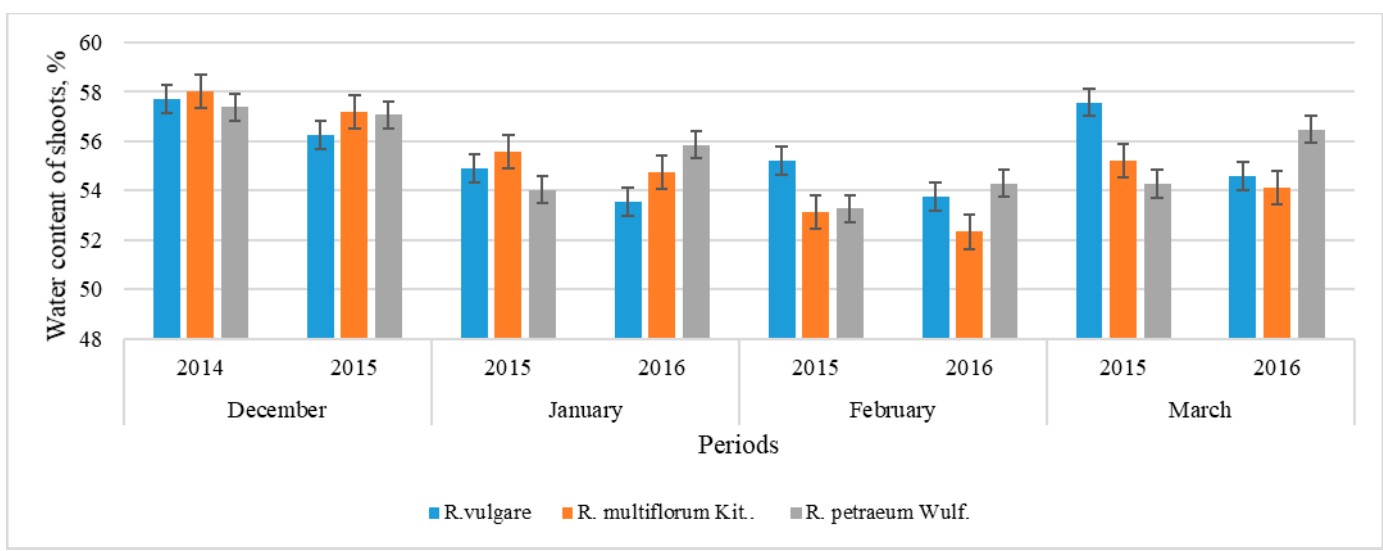

**Figure 3.** Changes in the water content in annual shoots of different red currant species during the winter months of 2014–2016. To compare the differences for each month and separately for each species, *t*-test was used.

During the winter months, there was a decrease in water in the tissues of annual shoots. By January, there was a decrease in the water content and an increase in the water content of shoots in all studied species of red currant, which was explained by their state of dormancy. Previous studies noted that hardy genotypes of fruit crops are characterized by a gradual decrease in hydration of shoots in winter [39,40]. In February, the medium and late-ripening cultivars maintained their resistance to temperature changes of the dormant period, and the hydration of annual shoots and water loss decreased. In the early maturation cultivars 'Niva','Asya' and 'Roza' their hydration increased significantly, while the water content decreased, which confirms the assumption of their early exit from the state of deep dormancy and is explained by genetic and ecological-geographical origin. These genotypes are descendants of Western European *R. vulgare* Lam. species with early ripening dates. At the beginning of spring (March), there were sharp jumps in the temperatures at night and during the day, which were the main reasons for the decrease in plant resistance. During this period, the water content of shoots indicator was especially relevant. In our experiments, damage to flower buds by negative temperatures and, as a consequence, low adaptation to negative environmental factors of the winter-spring period was noted in the representatives of *R. vulgare* Lam. ('Asya',' Niva', 'Roza') (Figure 4).

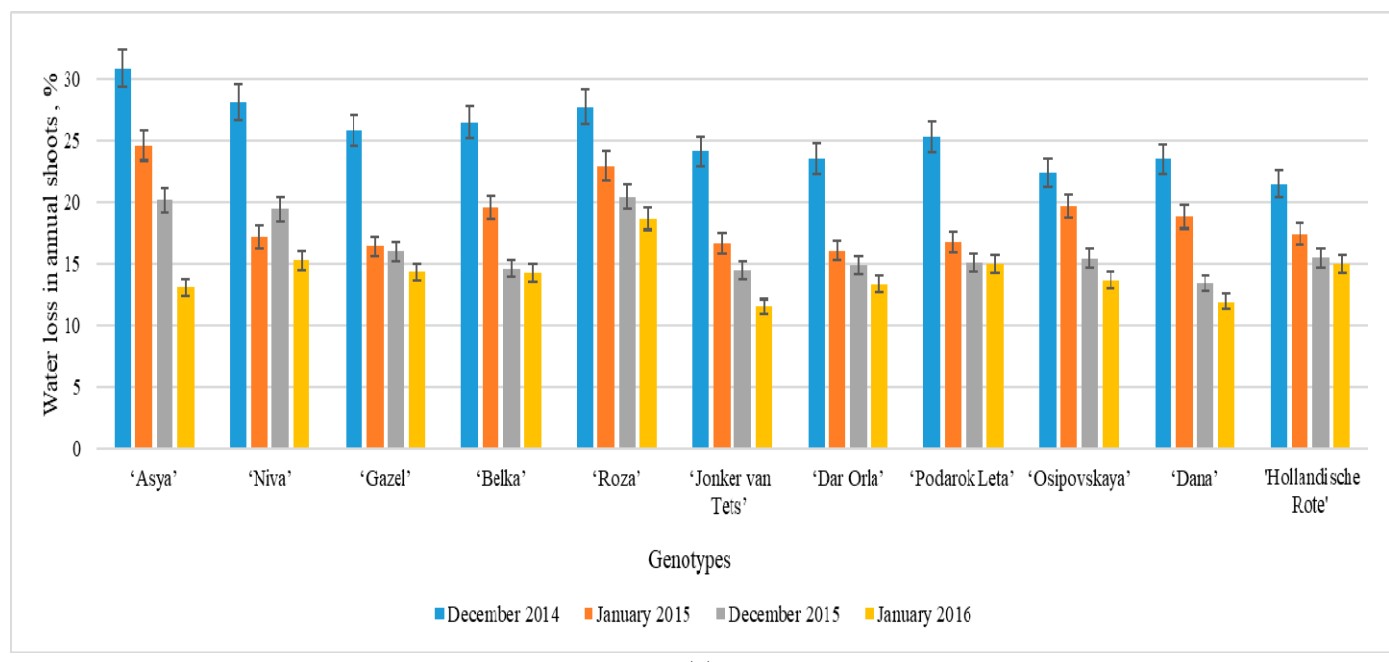

(**a**)

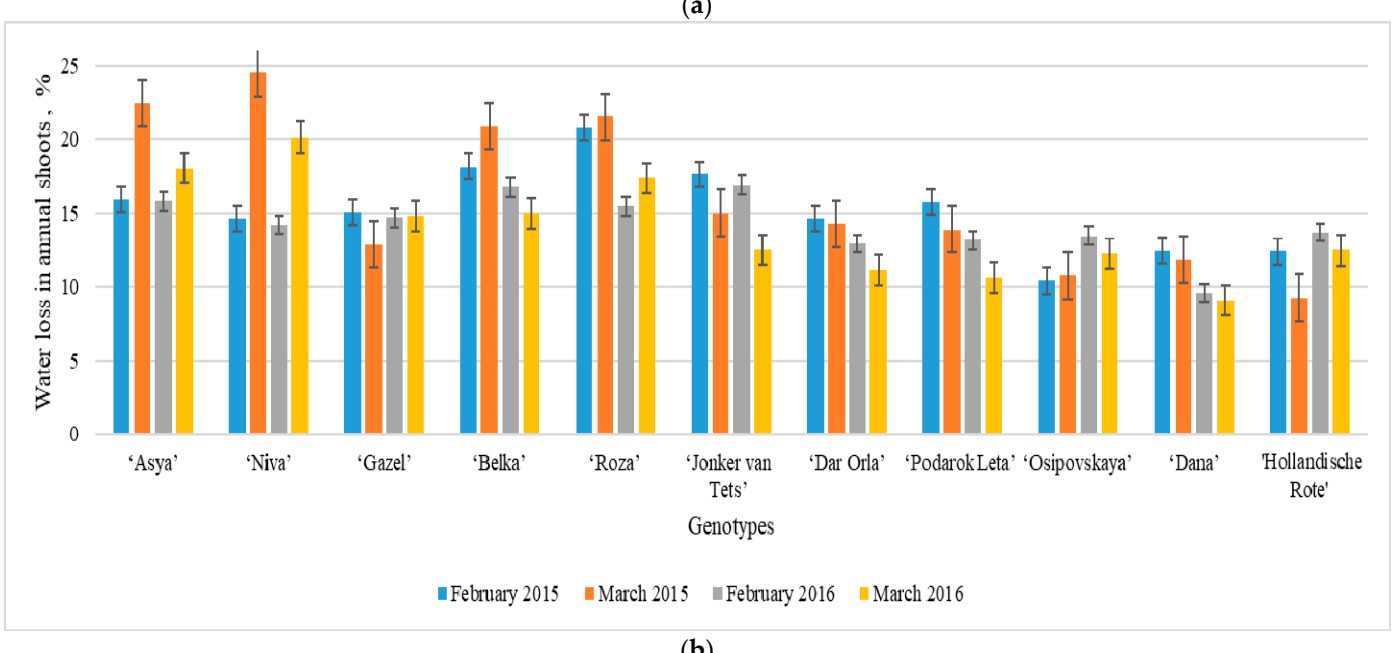

(**b**)

**Figure 4.** Laboratory assessment of the percentage of water loss by annual shoots of red currant cultivars at the beginning and middle of the winter period (**a**) and at the end of winter and early spring (**b**). To compare the differences for each month and separately for each species, *t*-test was used.

During the experiments, the results of the laboratory studies of the resistance of cultivars to low temperatures were compared with field observations. According to the results of the field observations, the cultivars of the early ripening period 'Asya' and 'Niva' as well as of the middle ripening period' Roza 'and' Belka', belonging to the species *R. vulgare* Lam., had a high percentage of damage to some flower buds and annual shoots, while perennial shoots were not significantly damaged by low temperatures (Figure 5).

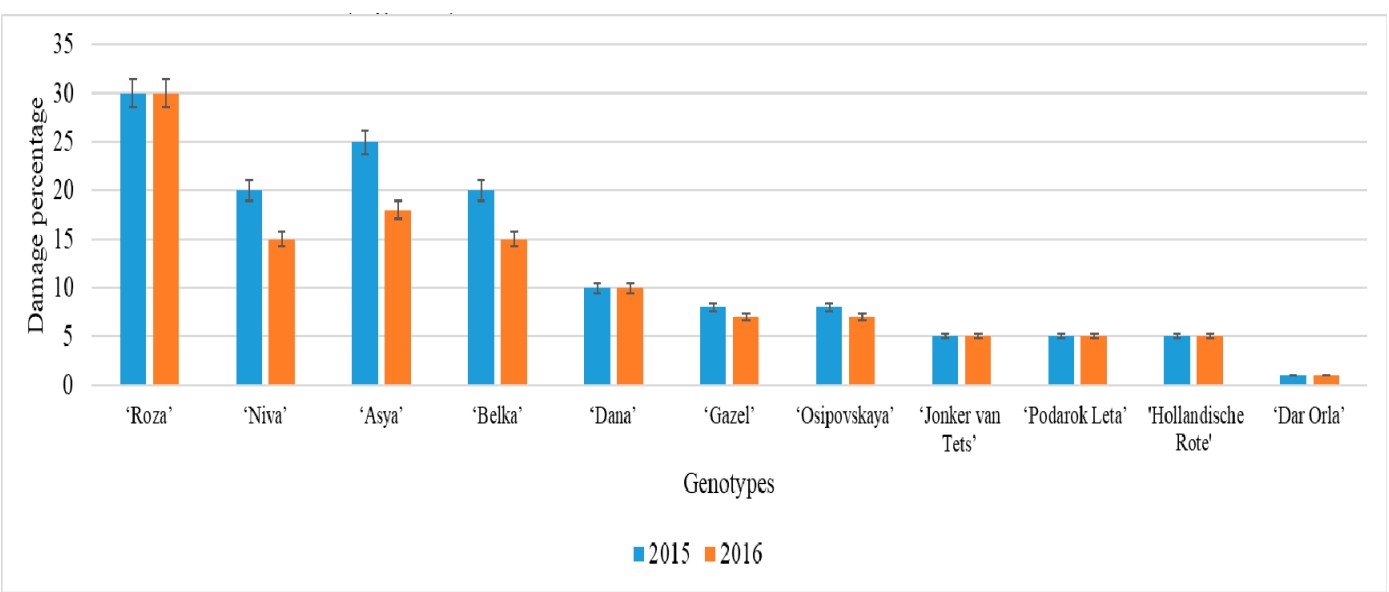

**Figure 5.** Field assessment of winter hardiness of the cultivars belonging to different species, damage to generative buds and annual shoots. To compare the differences between the study periods and a separate one for each genotype, *t*-test was used.

In general, after leaving the dormant state, the general condition of the plants of these cultivars was estimated as average: the bushes displayed weakened growth, with noticeable traces of winter damage. Generative buds had restrained development, the leaf blades were not aligned, the shoot-forming ability was low, the annual shoots were thin, and the growth and thickness were not aligned. Most VNIISPK breeding cultivars of early-medium and late maturation periods ('Osipovskaya', 'Podarok Leta', 'Gazel', 'Dar Orla' and 'Dana'), as well as the foreign breeding cultivars 'Jonkheer Van Tets' and 'Hollandische Rote' were quite winter-hardy, and showed minor damage. The apical part of the annual shoots froze (no more than $\frac{1}{4}$ of the length of the shoots), the perennial branches were not damaged by low temperatures, and in some years there were no signs of damage at all. For several years, these genotypes well tolerated the unfavorable winter conditions of the growing region. In general, the laboratory studies of the indicators of the water status during the dormant period of the plants were fully confirmed by the field observations and characterized the adaptive ability of the cultivar to abiotic stressors. The laboratory assessment of winter hardiness according to the indicators of the water status can be used as a criterion for the degree of suitability of the cultivar for cultivation in a certain region or zone. 'Jonkheer Van Tets', 'Hollandische Rote', 'Osipovskaya', 'Podarok Leta', 'Gazel', 'Dar Orla' and 'Dana' can be used in breeding programs of scientific institutions as a source of winter hardiness.

## 4. Conclusions

This study is aimed at finding measurements for quick and accurate prediction of the adaptive ability of red currant cultivars under certain environmental conditions, taking into account the species origin. The choice of red currant genotypes for introduction is determined by the genetic and geographical origin of the cultivars and the climate of the growing region. Cultivars belonging to different species, according to the terms of flowering and maturation, are divided into early, medium-early, medium and late ones. The extended period of fruiting of the cultivars of the *R. petraeum* Wulf. and *R. multiflorum* Kit. species allows consumers to use fresh berries of these cultivars for a fairly long period.

The dependence of ascorbic acid accumulation in berries on the species and temperature is shown, which makes it possible to determine the role of climatic factors in the synthesis of biologically active compounds in berry crops. The maximum accumulation of vitamin C in the cultivars belonging to *R. petraeum* Wulf. and *R. multiflorum* Kit. species is

determined by the temperature of +25 °C. The study shows the elements of physiological adaptation of cultivars to abiotic factors of the dormant period. Winter hardiness is determined by the water content in annual shoots and the degree of its consumption during the winter months. The indicators of the water status are confirmed by field observations and can be used as a criterion for assessing the winter hardiness of berry cultivars in the breeding programs of scientific institutions. The cultivars of the Western European species *R. vulgare* Lam. ('Asya', 'Niva', 'Roza') show a low physiological adaptive ability to abiotic stressors of the winter-spring period and are recommended for cultivation in the areas with a temperate continental climate.

Cultivars belonging to the species *R. petraeum* Wulf.-Hollandische Rote and R. multiflorum Kit., i.e., 'Osipovskaya', 'Podarok Leta', 'Gazel', 'Dar Orla' and 'Dana' are one the other hand characterized by high physiological resistance to low temperatures and can be used in breeding as sources and donors of winter hardiness and are recommended for cultivation in the areas with a continental climate.

**Author Contributions:** Conceptualization, writing—review and editing, O.P.; methodology, software, validation, M.T.; visualization, supervision, O.G.; project administration, funding acquisition S.K.; validation, M.K. All authors have read and agreed to the published version of the manuscript.

**Funding:** This research was funded by the Russian Ministry of Education and Science (Research No. 0467-2022-0001).

**Data Availability Statement:** In this study, nine red currant cultivars (*Ribesia* (*Berl.*) Jancz.) bred by the Russian Research Institute of Fruit Crop Breeding (VNIISPK) and two foreign bred cultivars (The Netherlands) were used. The genotypes were taken from the VNIISPK red currant cultivar study genetic collection site, and were kindly provided by Ph.D. Olga Golyaeva (Russian Research Institute of Fruit Crop Breeding, VNIISPK, Orel, Russia).

**Conflicts of Interest:** The authors declare no conflict of interest.

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
