# Peer review of "Agrometeorological and Morpho-Physiological Studies of the Response of Red Currant to Abiotic Stresses"

_agronomy, doi:10.3390/agronomy11081522_

Round 1

Reviewer 1 Report

As author and team are from non-English area so manuscript need to revise with a English expert before final submission. 

There are few suggestion by reviewer in attached file. Please find the attached file. 

Author Response

Dear reviewer,

we are correct manuscript.

All corrections are in red color.

Best 

regards,

Olga Panfilova

Reviewer 2 Report

This manuscript has good information that should be published. There are a few issues that should be addressed in a revision:

  • The English is rough in places. Some words are not commonly used in English, such as "sintering." Other words are not used correctly; e.g. "decade" (weeks, page 3?) and “optimal” (line 21). “Water-holding capacity” of shoots should probably just say “water content” of shoots. (Water-holding capacity is only used in relation to soils.) The manuscript would benefit from editing by a native English speaker.
  • Table 2. Most published data that relates plant growth phases to temperature use Growing Degree Days (GDD). My recommendation is to calculate GDDs starting from March 1 and then report the accumulated GDD when the varieties reached their particular growth stage (e.g. bud break, flowering, ripening). Then the data will be comparable with other such studies. The actual dates in Table 2 aren’t helpful without GDD information (which integrates time and temperature). The discussion from lines 191 – 216 should be revised based on GDD calculations.
  • In general, when reporting regression equations, the authors should also include the n= and r=. Otherwise, the reader doesn’t know how significant the regressions are.
  • In Figure 1, the axes are reversed. The independent variable (temperature) should always go on the x-axis and the dependent variable on the y-axis. This, of course, will change the equations.
  • Figures 2 and 3 – The labels are small and hard to read. The figure description needs to contain more information so it can stand alone. The temperatures on the axes do not need to be followed with .00. Removing the decimal points will help with the legibility.
  • An important component of the study is missing. The authors need to relate the measurements of water retention (and especially and loss in annual shoots) with the field observations of winter survival. Another table (or figure) is required that relates water loss to the field assessment of the various varieties. Otherwise, the reader doesn’t know if water loss is related to winter survival or if that measurement is a good predictor of winter survival.
  • Related to point #6, the visual assessment (lines 153-162) uses the word “frozen,” but I think the authors mean that the tissue was damaged to the point where the tissue was blackened. All of the plants experienced freezing conditions during winter. What the authors measured was tissue damage in response to cold temperatures. So to say “shoots and buds were not frozen” is inaccurate.

Author Response

Уважаемый рецензент, это верная рукопись. Все исправления отмечены синим цветом.

Лучший

С уважением,

Ольга Панфилова

Round 2

Reviewer 2 Report

The title seems too complex and not descriptive. I suggest, "Morphological and physiological measurements of red currant cultivars of various origins predict adaptability to a cold climate."

The accepted genus for red currants is Ribes rubrum L. from what I understand. Ribesia is not a recognized genus.

There are still some unusual uses of the English language. Here are a few: 

line 13 - "methods" should be "measurements"

line 16, 56 - "regime" should be "status"

line 31 - "decade" is correct, not "time"

 line 38 - remove "culture"

line 41 - remove "allows to increase" and replace with "increases"

line 60 - should be "content"

There are others, but these are examples.

line 32-33 - Could a lower price for consumers also be responsible for increasing demand? I think this is probably more important than a perception that the food is healthy.

line 103 - need to provide the extractant used for the soil test - this determines the amount of P and K extracted from the soil solution

Using Growing Degree Days has improved the paper significantly